# From Arithmetic to Logic: The Resilience of Logic and Lookup-Based Neural Networks Under Parameter Bit-Flips

**Alan T. L. Bacellar**                             *alanbacellar@utexas.edu*
**Sathvik Chemudupati**                               *sathvikc@utexas.edu*
**Shashank Nag**                                   *shashanknag@utexas.edu*
**Allison Seigler**                                   *aseigler@utexas.edu*
*The University of Texas at Austin*

**Priscila M. V. Lima**                             *priscilamvl@cos.ufrj.br*
*Federal University of Rio de Janeiro*

**Felipe M. G. França**                                *felipe@ieee.org*
*Instituto de Telecomunicações, Porto, Portugal (now at Google LLC)*

**Lizy K. John**                                    *ljohn@ece.utexas.edu*
*The University of Texas at Austin*

**Reviewed on OpenReview:** *https://openreview.net/forum?id=ZZYvGZei5h*

## Abstract

The deployment of Deep Neural Networks (DNNs) in safety-critical edge environments necessitates robustness against hardware-induced bit-flip errors. While empirical studies indicate that reducing Numerical Precision can improve fault tolerance, the theoretical basis of this phenomenon remains underexplored. In this work, we study resilience as a Structural Property of neural architectures rather than solely as a property of a dataset-specific trained solution. By deriving the Expected Squared Error (MSE) under independent parameter bit flips across multiple numerical formats and layer primitives, we show that lower precision, higher sparsity, bounded activations, and shallow depth are consistently favored under this corruption model. We then argue that Logic and Lookup-Based Neural Networks realize the joint limit of these design trends. Through ablation studies on the MLPerf Tiny benchmark suite, we show that the observed empirical trends are consistent with the theoretical predictions, and that LUT-based models remain highly stable in corruption regimes where standard floating-point models fail sharply. Furthermore, we identify a novel Even-Layer Recovery effect unique to logic-based architectures and analyze the structural conditions under which it emerges. Overall, our results suggest that shifting from continuous arithmetic weights to discrete Boolean lookups can provide a favorable Accuracy–Resilience trade-off for hardware fault tolerance.

## 1 Introduction

The ubiquitous deployment of Deep Neural Networks (DNNs) in safety-critical edge environments—from autonomous vehicles to implantable medical devices—has exposed a critical vulnerability: hardware reliability. These edge environments are often hostile, characterized by ionizing radiation, thermal fluctuations, and aggressive voltage scaling, all of which compromise the physical integrity of memory and logic (Hochschild et al., 2021; Wang et al., 2023). While server-grade systems can rely on expensive Reliability, Availability, and Serviceability (RAS) mechanisms like Chipkill ECC to mask these faults (Mukherjee, 2011), power-constrained edge accelerators often lack the energy and silicon budget for such redundancy. Consequently, the burden of resilience has shifted from the underlying hardware to the neural architecture itself.

Historically, intuition grounded in classical information theory suggested that high-precision formats, such as FP32, offered the greatest robustness due to their wider dynamic range and representational redundancy. However, a growing body of empirical evidence has dismantled this assumption. Recent studies indicate that reducing numerical precision paradoxically improves fault tolerance, with Binary Neural Networks (BNNs) sustaining accuracy under error rates that render full-precision models useless (Ghavami et al., 2024). This phenomenon suggests that the discretization of the parameter space does not merely compress the model but acts as a powerful structural regularizer against noise.

Prior attempts to explain this phenomenon have typically relied on analyzing the loss landscape, arguing that quantized models converge to "flatter" minima (Baldi et al., 2025), or have depended on purely empirical fault injection campaigns on specific datasets. However, these explanations are inherently dataset-dependent; a flat minimum found for one dataset does not guarantee robustness on another. Such analyses characterize the *trained state* of the model rather than the *structural resiliency* of the neural architecture itself.

In this work, we posit that these observations are not isolated anomalies but reflect a common structural trend in bit-flip resilience. We identify four architectural factors that consistently improve resilience under our corruption model: lower precision, higher sparsity, bounded activations, and shallower compositions. We further argue that Logic and Lookup-Based Neural Networks (LUT-NNs), also known as Weightless Neural Networks (WNNs) (Aleksander et al., 1984; 2009b; Bacellar et al., 2024b), realize an extreme point in this design space. By replacing numerical weights with discrete truth tables, enforcing localized sparse connectivity, and using inherently bounded Boolean outputs, these models exhibit a qualitatively different fault-propagation mechanism from standard weighted networks.

To study this thesis, we derive a formal, dataset-agnostic framework for the expected squared output error induced by parameter bit flips. Our analysis isolates the role of numerical representation and architectural structure, and yields explicit neuron- and layer-level error expressions for several common formats and operations. Within this framework, floating-point representations admit rare but potentially very large multiplicative distortions through exponent corruption, whereas lookup-table neurons localize the effect of a parameter fault to a bounded subset of input addresses.

We present a theoretical and empirical study of this resilience landscape. Our main contributions are:

1. **Neuron- and layer-level error analysis:** We derive expected output error expressions under independent parameter bit flips for integer, floating-point, quantized, binary, and LUT-based neuron models.

2. **Structural resilience trends:** We show, within this analytical framework, how lower precision, bounded activations, sparsity, and shallow compositions improve resilience under bit-flip corruption.

3. **Ablation-driven validation:** Through experiments on the MLPerf Tiny benchmark suite (Banbury et al., 2021), we isolate the impact of width, depth, precision, activation, and sparsity, and show that the empirical trends are consistent with the theory.

4. **Logic/LUT robustness:** We compare LUT-based architectures against weighted baselines under severe corruption and show that Logic/LUT models remain stable substantially deeper into the fault regime.

5. **Symmetric recovery:** We analyze a distinctive Even-Layer Recovery effect that emerges in logic-based networks at extreme fault rates.

## 2 Related Work

**Hardware Faults, Silent Data Corruptions, and Attacks.** The susceptibility of Deep Neural Networks to hardware-induced faults has been extensively documented, prompting fundamental analyses of DNN resiliency and the development of analytical frameworks to track uncertainty propagation through network layers (Jungmann et al., 2024). However, these analytical models traditionally focus on continuous-valued parameter perturbations and assume infinitesimal errors, limiting their ability to model the large-magnitude,

discrete disruptions caused by hardware bit-flips. To quantify these vulnerabilities at an industrial scale, the Parameter Vulnerability Factor (PVF) metric was proposed to evaluate the impact of silent data corruptions (SDCs) on production models (Jiao et al., 2024). Beyond natural faults, attackers leverage techniques like Progressive Bit Search to crush network accuracy through targeted malicious bit-flips (Rakin et al., 2019), a vulnerability that remains a critical threat to modern Large Language Models (Chen et al., 2024). Recent discoveries reveal that these attack surfaces extend directly into compiled DNN executables (Chen et al., 2025b), requiring specialized defense frameworks like BITSHIELD to secure compiled code (Chen et al., 2025c). To mitigate errors at the model level, researchers have proposed defending through weight reconstruction (Li et al., 2020), exploiting bit-level redundancy (Catalán et al., 2024), and enforcing bit homogeneity (MONO) within parameters (Eslami et al., 2024).

**Quantization, Resilient Training, and Vulnerability.** Reducing numerical precision—a foundational technique for efficient inference (Jacob et al., 2018; Hubara et al., 2017; Qin, 2020; Pouransari et al., 2020; Zahran et al., 2025)—has demonstrated empirical robustness benefits. These observations are supported by loss landscape analyses designed for reliable quantized models (Baldi et al., 2025), motivating the development of inherently resilient Binary Neural Networks (BNNs) (Xu et al., 2023) and methods for learning fault-resistant quantization ranges (Chitsaz et al., 2023). While Sen et al. (2020) show that mixed-precision ensembles improve robustness against input-space adversarial attacks, their focus remains on input perturbations rather than memory corruption or BER-controlled bit-flips. Despite these benefits, quantized models remain vulnerable, necessitating formal verification of bit-flip attacks (Lin et al., 2025), investigations into BNN adversarial weaknesses (Kundu et al., 2024), and defenses against scale-factor manipulation (Wang et al., 2025). Complementarily, Chen et al. (2025a) demonstrate that lower-bit PTQ/QAT can degrade LLM alignment safety, yet their work examines prompt-induced degradation rather than the weight-level memory corruption we study.

**Logic and LUT-Based Architectures.** Moving beyond traditional arithmetic representations to mitigate these propagating errors, Logic and LUT-Based Neural Networks synthesize neural operations into pure Boolean logic and discrete memory lookups. One approach compiles sparse, quantized neurons directly into lookup tables for extreme-throughput applications, as exemplified by LogicNets (Umuroglu et al., 2020) and FPGA implementations like NeuraLUT (Andronic & Constantinides, 2024). Recent advancements have expanded this paradigm through DiffLogicNets, enabling the fully differentiable training of logic gates (Petersen et al., 2022). A closely related approach relies natively on discrete memory lookups (RAM nodes) rather than compiling trained arithmetic weights into logic (Aleksander et al., 2009a). Instead of learning arithmetic parameters, these models directly learn the Boolean contents of the memory addresses. This purely discrete architecture has recently seen a resurgence for resource-constrained environments. Crucially, the DWN architecture advanced this paradigm by enabling the direct, gradient-based training of these discrete LUT entries (Bacellar et al., 2024b), providing a highly efficient alternative to standard backpropagation. Building on these discrete LUT-based concepts, hybrid approaches have demonstrated the practical viability of combining RAM-nodes with standard arithmetic operations for edge inference (Jadhao et al., 2024), while LL-ViT has successfully adapted these logic and lookup-based principles to state-of-the-art Vision Transformers (Nag et al., 2025). While these works highlight the immense computational efficiency of logic and memory lookups, our work uniquely bridges the gap by formalizing their structural capacity for extreme fault resilience.

## 3 Background

### 3.1 Logic and Lookup-Based Neural Networks

Recent advances in neural network design for ultra-low-power edge environments have proposed replacing traditional arithmetic multiply-accumulate (MAC) operations with pure Boolean logic. Early approaches in this domain, such as Differentiable Logic Gate Networks (DiffLogicNets) (Petersen et al., 2022), construct multi-layer networks entirely from randomly wired binary logic gates (e.g., AND, OR, XOR).

However, pure logic gate networks represent a constrained subset of a much broader architectural space: Lookup Table (LUT) based models. A LUT is essentially a discrete memory array that maps an input

address to an output value. A simple LUT with 2 binary inputs contains $2^2 = 4$ memory bits and can theoretically represent all 16 possible 2-input Boolean logic gates. Consequently, LUT-based architectures naturally generalize pure logic networks. By expanding the fan-in to $K$ inputs, a single LUT can express any of the $2^{2^K}$ possible Boolean functions for its input variables. This exponentially expands the hypothesis space and expressivity of the neural node while completely bypassing arithmetic operations during inference. Beyond their theoretical expressivity, these architectures provide highly motivated practical advantages; they have demonstrated substantial improvements in inference efficiency over standard Binary Neural Networks (BNNs), achieving up to a 2000× reduction in area-delay product (Bacellar et al., 2024b).

### 3.2 Differentiable Weightless Neural Networks (DWNs)

In this work, we utilize Differentiable Weightless Neural Networks (DWNs) (Bacellar et al., 2024b) as the representative architecture for logic and LUT-based models. DWNs fully leverage the generalization power of multi-input LUTs and enable end-to-end gradient-based optimization without relying on arithmetic proxy weights during inference.

**Architectural Structure.** The DWN architecture replaces numerical weight matrices with layers of trainable LUTs. Figure 1 shows an example of a simple 2-layer DWN model.

- **Input Mapping:** A DWN layer receives a binary input vector $X$. Each LUT in the layer is assigned $K$ inputs (the fan-in). These $K$ inputs are sampled from $X$ using a fixed, pseudo-random connectivity matrix.

- **Lookup Operation:** The $K$ sampled binary values are concatenated to form an integer address $a \in \{0, \ldots, 2^K - 1\}$. The LUT performs a direct memory access, returning the binary value stored at index $a$ in its internal truth table $T \in \{0, 1\}^{2^K}$.

- **Connectivity:** The binary outputs of the LUTs in layer $l$ are sparsely connected to form the inputs for layer $l + 1$.

- **Classification Head:** In the final layer, the network produces continuous logits by aggregating the binary outputs of the terminal LUTs. This is typically done using a simple, hardware-friendly popcount (summation) operation, counting the total number of "votes" for each specific class.

**Differentiable Training on Discrete Memory.** Because the lookup operation is a discrete addressing step rather than a continuous mathematical function, it naturally blocks the backpropagation of gradients. While older models relied on one-layer models (Aleksander et al., 1984; Susskind et al., 2023; Bacellar et al., 2024a), DWNs overcome this limitation by introducing the Extended Finite Difference (EFD) method. During the backward pass, EFD estimates the gradient by analytically approximating how a discrete bit-flip in a specific LUT memory entry $T[a]$ would influence the continuous loss function at the output.

This formulation allows the contents of the discrete truth tables to act as the direct, learnable parameters of the network. By treating the LUT entry itself as the parameter optimized by stochastic gradient descent, DWNs align the training objective directly with the resilient, logic-based substrate. This makes DWNs an ideal candidate for our analysis, as they represent the convergence of minimal precision (1-bit storage), hard saturation (Boolean outputs), and high sparsity (fixed $K$-input fan-in).

**Expressivity, efficiency, and scalability motivation.** Logic and lookup-based networks are not merely small special-purpose classifiers. A $K$-input LUT can represent any Boolean function over its local input variables, and compositions of many LUTs form a high-dimensional discrete hypothesis space. Prior work on VC-dimension analysis shows that 1-layer LUT-based models can have substantial capacity and scalable expressivity (Carneiro et al., 2019). This suggests that, at least over binary or discretized representations, LUT networks can serve as a general function-approximation substrate rather than only as hand-crafted logic. Recent logic/LUT-based systems further show that this substrate can deliver extremely low-energy, low-latency, and high-throughput inference, especially for MLP-style implementations or hybrid architectures that combine LUT/RAM nodes with weighted components (Umuroglu et al., 2020; Petersen et al., 2022;

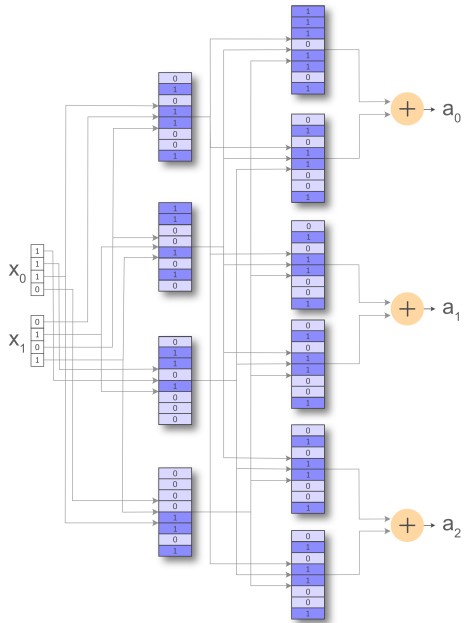

Figure 1: Illustration of a 2-layer neural network composed entirely of layers of lookup tables with binary values. The outputs of each layer address the LUTs of the next layer, culminating in a popcount for class activations.

Andronic & Constantinides, 2024; Bacellar et al., 2024b; Jadhao et al., 2024; Petersen et al., 2024; Nag et al., 2025; Andronic & Constantinides, 2025; Weng et al., 2025; Khataei & Bazargan, 2025; Lou et al., 2025; Cassidy et al., 2025; Hoang et al., 2026; Rüttgers et al., 2026; Gerlach et al., 2026; Kresse & Lampert, 2026). However, current results do not yet show that large CNNs or Transformers can be replaced end-to-end by purely LUT-based models without additional architectural advances. Our work adds a complementary motivation for studying this direction: logic and lookup-based models are attractive not only because they can yield extreme inference efficiency, but also because they provide extreme structural resilience to parameter bit flips. This motivates deeper research into scaling these models toward fully convolutional, transformer, and foundation-model settings.

# 4 Theoretical Development

In this section, we derive the expected squared error induced by random bit-flip errors in the memory cells storing neural network weights and parameters. Rather than attempting to characterize resilience only through a particular trained model or dataset, we isolate the structural sources of corruption sensitivity: numerical representation, fan-in, sparsity, activation boundedness, and depth. The resulting expressions are local in the sense that they describe the corruption-induced output drift of individual numerical formats, neurons, or layer primitives. These local quantities are nevertheless the building blocks of network-level resilience. We therefore use a modular analysis: first deriving how each architectural factor affects expected output drift, and then studying how these errors compose through width, sparsity, nonlinearities, and depth. In particular, the depth model in Section 4.7 provides a simple composition mechanism showing when layer-level corruption errors attenuate, accumulate approximately linearly, or amplify across the full network.

## 4.1 Definitions and Noise Model

Let a DNN neuron pre-activation computation be defined as $y = \sum_{i=1}^{n} w_i x_i$, where $x \in \mathbb{R}^n$ is the input vector and $w \in \mathbb{R}^n$ is the weight vector. We assume a Bit Error Rate (BER) $p$, where each bit in the binary representation of the parameters flips independently with probability $p$ (i.e., independent Bernoulli trials).

Let $w'$ denote the corrupted weight and $y'$ the corrupted output. The objective is to find the expected squared error (MSE), decomposed into bias and variance:

$$E[(y' - y)^2] = \text{Var}(y') + (E[y'] - y)^2 \tag{1}$$

**Scope of the independent-bit model.** The independent Bernoulli model should be read as a first-order abstraction of random silent data corruption rather than a complete hardware fault model. It is most appropriate for isolated memory upsets, or for systems where physical layout, interleaving, or protection mechanisms decorrelate faults across logical parameter bits. Real hardware can also produce correlated faults, such as stuck-at cells, word-line failures, burst errors, or multi-bit upsets. In those cases, the closed-form expressions below would acquire covariance terms and the probability of shared layer-wide distortion can be larger than predicted by independent flips. The qualitative structural implications remain the same when faults are local and bounded—lower precision, sparsity, bounded activations, and LUT-localized addressing reduce the number and magnitude of affected computations—but our numerical MSE expressions should not be interpreted as exact predictions for strongly correlated hardware failures. Evaluating such structured fault models is an important extension.

## 4.2 Weight Representations

### 4.2.1 Integer Weights

We first consider weights represented as $B$-bit integers in Two's Complement format. A weight $w$ is given by $w = -b_{B-1}2^{B-1} + \sum_{k=0}^{B-2} b_k 2^k$.

**Theorem 1** (Expected Error for Integer Weights). *For a neuron with fan-in $n$, inputs $x$, and $B$-bit integer weights $w$ subject to independent bit flips with probability $p$, the expected squared error is:*

$$E[(y' - y)^2] = \underbrace{p(1 - p)\left(\frac{4^B - 1}{3}\right)\|x\|_2^2}_{\textit{Variance Term}} + \underbrace{p^2 \left(\sum_{i=1}^{n} x_i(1 + 2w_i)\right)^2}_{\textit{Bias Term}} \tag{2}$$

*Proof.* Let $\epsilon_k$ be the error introduced by a flip in bit $k$. The total weight error is $\Delta w = \sum_{k=0}^{B-1} \epsilon_k$. The expectation of the error for bit $k$ with value $b_k$ and place value $V_k$ is $E[\epsilon_k] = p(1 - 2b_k)V_k$. Summing over all bits, the mean weight error is:

$$E[\Delta w] = p \sum_{k=0}^{B-1}(1 - 2b_k)V_k = p\left(\sum V_k - 2w\right) = p(-1 - 2w) \tag{3}$$

where we use the identity $\sum_{k=0}^{B-1} V_k = -1$ for Two's Complement.

The variance of the error is the sum of independent bit variances. For a single bit, $\text{Var}(\epsilon_k) = p(1 - p)V_k^2$. Summing over $k$:

$$\text{Var}(\Delta w) = p(1 - p) \sum_{k=0}^{B-1} V_k^2 = p(1 - p)\frac{4^B - 1}{3} \tag{4}$$

The output error $\Delta y = \sum x_i \Delta w_i$ is a sum of independent variables. Thus, $\text{Var}(\Delta y) = \sum x_i^2 \text{Var}(\Delta w_i)$ and $E[\Delta y] = \sum x_i E[\Delta w_i]$. Substituting these into the bias-variance decomposition $E[\Delta y^2] = \text{Var}(\Delta y) + E[\Delta y]^2$ yields the theorem. $\square$

### 4.2.2 Floating-Point Weights

We assume a floating-point representation with one sign bit $s$, $E$ exponent bits, and $M$ mantissa bits:

$$w = (-1)^s \cdot m \cdot 2^{e-\text{bias}}, \tag{5}$$

where $m = 1 + \sum_{k=1}^{M} f_k 2^{-k}$ and $e = \sum_{j=0}^{E-1} b_j 2^j$. Unlike integer formats, bit flips in floating-point may induce multiplicative distortions through the exponent.

For analytical tractability, we consider the finite decoded outcomes of the corrupted representation. Subnormal values are included in the moments below, while NaNs and $\pm\infty$ are excluded from the moment calculation.

**Theorem 2** (Expected Error for Floating-Point Weights). *Let $w_i'$ denote the corrupted floating-point weights after independent bit flips with probability p, and define*

$$\Gamma_i = \mathbb{E}[w_i'], \qquad \Omega_i = \mathbb{E}[w_i'^2]. \tag{6}$$

*Then*

$$\mathbb{E}[(y' - y)^2] = \sum_{i=1}^{n} x_i^2 \left(\Omega_i - \Gamma_i^2\right) + \left(\sum_{i=1}^{n} x_i(\Gamma_i - w_i)\right)^2. \tag{7}$$

*Proof.* Since $y' = \sum_{i=1}^{n} x_i w_i'$, independence across corrupted weights gives

$$\mathrm{Var}(y') = \sum_{i=1}^{n} x_i^2 \mathrm{Var}(w_i') = \sum_{i=1}^{n} x_i^2 (\Omega_i - \Gamma_i^2), \tag{8}$$

and

$$\mathbb{E}[y'] - y = \sum_{i=1}^{n} x_i(\Gamma_i - w_i). \tag{9}$$

Applying the bias–variance decomposition yields the claim. $\square$

Write

$$w_i' = S_i' \, Man_i' \, Exp_i', \tag{10}$$

with

$$S_i' = (-1)^{s_i'}, \qquad Man_i' = 1 + \sum_{k=1}^{M} f_{ik}' 2^{-k}, \qquad Exp_i' = 2^{e_i' - \mathrm{bias}}. \tag{11}$$

Under independent bit flips,

$$\Gamma_i = \mathbb{E}[S_i'] \, \mathbb{E}[Man_i'] \, \mathbb{E}[Exp_i'], \tag{12}$$

$$\Omega_i = \mathbb{E}[S_i'^2] \, \mathbb{E}[Man_i'^2] \, \mathbb{E}[Exp_i'^2]. \tag{13}$$

The sign moments are

$$\mathbb{E}[S_i'] = (1 - 2p)(-1)^{s_i}, \qquad \mathbb{E}[S_i'^2] = 1. \tag{14}$$

The mantissa behaves as a bounded fixed-point quantity, with

$$\mathbb{E}[Man_i'] = Man_i + p \sum_{k=1}^{M} (1 - 2f_{ik}) 2^{-k}, \tag{15}$$

and

$$\mathbb{E}[Man_i'^2] = \mathbb{E}[Man_i']^2 + p(1-p) \sum_{k=1}^{M} 4^{-k}. \tag{16}$$

Hence the mantissa contribution remains uniformly bounded.

For the exponent,

$$\mathbb{E}[Exp_i'] = 2^{e_i - \mathrm{bias}} \prod_{j=0}^{E-1} \left[(1-p) + p \, 2^{(1-2b_{ij})2^j}\right], \tag{17}$$

and

$$\mathbb{E}[Exp_i'^2] = 2^{2(e_i - \text{bias})} \prod_{j=0}^{E-1} \left[ (1-p) + p\, 2^{2(1-2b_{ij})2^j} \right]. \tag{18}$$

Thus exponent corruption dominates the floating-point error: while mantissa perturbations remain bounded, a single flipped exponent bit contributes factors of the form $2^{\pm 2^j}$. Accordingly, the second moment admits worst-case doubly-exponential growth in the exponent width,

$$\mathbb{E}[Exp_i'^2] = O\left( 2^{2^E} \right), \tag{19}$$

up to format-dependent constants. We use this as a worst-case upper-bound statement rather than a generic scaling law for every weight.

### 4.2.3 Affine Quantized (AQ) Weights

Pure integer quantization is highly resilient to bit flips, but in practice it often does not achieve the same accuracy as affine quantization because it cannot fully capture the scale and offset of real-valued weight distributions. For this reason, practical quantized models usually reconstruct real weights as

$$\hat{w} = S(w - Z), \tag{20}$$

where $w$ is the stored integer weight, $Z$ is the integer zero-point, and $S$ is the floating-point scale.

Affine quantization can be applied at different granularities: model-wise, layer-wise, or weight-/neuron-wise. Finer granularity generally gives better accuracy, with weight-/neuron-wise usually best and model-wise worst. However, finer granularity also increases storage overhead because more quantization parameters must be stored. In the extreme, per-weight affine quantization can nearly triple storage once a separate scale and zero-point must be stored for each weight. For this reason, layer-wise affine quantization is often the practical choice.

This granularity also changes the fault model. In pure integer quantization, a bit flip affects only one stored weight. In layer-wise affine quantization, a bit flip in an integer weight still causes a local error, but a bit flip in the shared quantization parameters affects all reconstructed weights in the layer. In particular, corruption in the integer zero-point $Z$ introduces a shared additive distortion, while corruption in the floating-point scale $S$ introduces a shared multiplicative distortion. Thus, affine quantization introduces a rare but potentially catastrophic layer-wide failure mode.

At the same time, such events are statistically unlikely. For a layer with $n$ quantized weights of $B_w$ bits each, integer zero-point precision $B_Z$, and floating-point scale precision $B_S$, the total number of bits is

$$N_{\text{tot}} = nB_w + B_Z + B_S. \tag{21}$$

Hence,

$$P(\text{quant. param. bit}) = \frac{B_Z + B_S}{nB_w + B_Z + B_S}, \tag{22}$$

which is much smaller than the probability of hitting an ordinary weight bit when $n \gg 1$. This quantity is not the output error itself. It is the fraction of stored layer bits belonging to the shared affine quantization parameters (metadata), equivalently the probability that a uniformly random corrupted bit hits the scale or zero-point rather than an ordinary integer weight. We use it to separate two effects: frequent localized integer-weight errors and rare shared metadata errors. When such a metadata fault occurs, the resulting output error is characterized by the two cases below: protected quantization parameters recover the localized integer behavior, while corrupted $Z$ and $S$ introduce additive and multiplicative layer-wide distortions. Therefore, under random independent bit flips, most faults still strike the quantized weights rather than the shared quantization parameters.

We therefore focus on the practically relevant layer-wise setting and consider two cases.

**Case 1: Protected Quantization Parameters**

**Theorem 3** (Layer-wise AQ Error with Protected Quantization Parameters). *Let*

$$y = S \sum_{i=1}^{n} x_i(w_i - Z), \qquad y' = S \sum_{i=1}^{n} x_i(w_i' - Z), \tag{23}$$

*where $w_i'$ are corrupted integer weights, while the shared floating-point scale $S$ and integer zero-point $Z$ are fault-free. Then*

$$\mathbb{E}[(y' - y)^2] = S^2 \sum_{i=1}^{n} x_i^2 \mathrm{Var}(w_i') + S^2 \left( \sum_{i=1}^{n} x_i(\mathbb{E}[w_i'] - w_i) \right)^2. \tag{24}$$

*Proof.* Since $S$ and $Z$ are unchanged,

$$y' - y = S \sum_{i=1}^{n} x_i(w_i' - w_i). \tag{25}$$

Hence

$$\mathbb{E}[(y' - y)^2] = \mathrm{Var}(y') + (\mathbb{E}[y'] - y)^2. \tag{26}$$

Using independence across corrupted weights,

$$\mathrm{Var}(y') = S^2 \sum_{i=1}^{n} x_i^2 \mathrm{Var}(w_i'), \tag{27}$$

and

$$\mathbb{E}[y'] - y = S \sum_{i=1}^{n} x_i(\mathbb{E}[w_i'] - w_i). \tag{28}$$

Substituting yields the result. $\square$

Thus, when the quantization parameters are protected, affine quantization preserves the bounded and localized error behavior of integer weights, scaled by the fixed factor $S^2$.

**Case 2: Corrupted Quantization Parameters**

**Theorem 4** (Structure of Layer-wise AQ Error with Corrupted Quantization Parameters). *Let*

$$y' = S' \sum_{i=1}^{n} x_i(w_i' - Z'), \tag{29}$$

*where the integer weights $w_i'$, the floating-point scale $S'$, and the integer zero-point $Z'$ may all be corrupted. Then the output error contains both a localized integer term and shared layer-wide terms, with*

$$\mathrm{Var}\left( \sum_{i=1}^{n} x_i(w_i' - Z') \right) = \sum_{i=1}^{n} x_i^2 \mathrm{Var}(w_i') + \left( \sum_{i=1}^{n} x_i \right)^2 \mathrm{Var}(Z'). \tag{30}$$

*Moreover, corruption in the floating-point scale $S'$ multiplicatively scales the entire accumulation, introducing an additional shared distortion analogous to the floating-point failure mode.*

*Proof.* Define

$$H' = \sum_{i=1}^{n} x_i(w_i' - Z') = \sum_{i=1}^{n} x_i w_i' - Z' \sum_{i=1}^{n} x_i. \tag{31}$$

Assuming the corrupted weights are independent across $i$ and independent of $Z'$, the variance decomposes as

$$\text{Var}(H') = \text{Var}\left(\sum_{i=1}^{n} x_i w_i'\right) + \text{Var}\left(Z' \sum_{i=1}^{n} x_i\right). \tag{32}$$

Therefore,

$$\text{Var}(H') = \sum_{i=1}^{n} x_i^2 \text{Var}(w_i') + \left(\sum_{i=1}^{n} x_i\right)^2 \text{Var}(Z'). \tag{33}$$

The first term is the usual localized contribution from corrupted integer weights. The second term is a correlated layer-wide contribution caused by the shared zero-point. Since $y' = S'H'$, corruption in the floating-point scale $S'$ additionally scales the full accumulation, producing another layer-wide distortion. $\square$

Therefore, layer-wise affine quantization is generally less resilient than pure integer quantization, because it introduces a small-probability but high-impact failure mode through the shared quantization parameters. However, it remains more resilient than floating-point weights in expectation, since most random bit flips still strike bounded integer payloads rather than floating-point exponent fields.

### 4.3 Binary Neural Networks (BNNs)

Binary Neural Networks represent the limit of quantization where weights are restricted to $w \in \{-1, +1\}$. This is typically implemented using a sign function on latent weights during training (Bengio et al., 2013; Yin et al., 2019; Hubara et al., 2016), leading to direct 1-bit storage $(0 \rightarrow -1, 1 \rightarrow +1)$ at inference time.

#### 4.3.1 Bit-Flip Error in BNNs

Let the mapping be $w = 2b - 1$ where $b \in \{0, 1\}$. A bit flip $(b \rightarrow 1 - b)$ causes the weight to flip sign completely $(w \rightarrow -w)$. The magnitude of the error is always fixed at $|\Delta w| = 2$.

**Theorem 5** (Expected MSE for BNNs). *For a BNN layer with inputs $x$ and binary weights $w$, subject to bit flips with probability $p$:*

$$E[(y' - y)^2] = 4p(1 - p)\|x\|_2^2 + 4p^2 \left(\sum_{i=1}^{n} x_i w_i\right)^2 \tag{34}$$

*Proof.* The error in a single weight is $\Delta w$. If a flip occurs (probability $p$), $\Delta w = -2w$. Mean error: $E[\Delta w] = p(-2w) = -2pw$. Variance: $\text{Var}(\Delta w) = E[\Delta w^2] - E[\Delta w]^2 = p(4) - (-2pw)^2 = 4p - 4p^2 = 4p(1-p)$. Substituting these into the general bias-variance equation yields the result. $\square$

**Scaling Analysis:** Unlike multi-bit integers where error magnitude depends on *which* bit flips (Sign vs LSB), BNN errors are uniform. While the variance coefficient ($4p$) is high compared to the LSB of an INT8 weight, BNNs have no "explosive" failure modes (like MSB flips in integers or exponent flips in floats). They are maximally robust in terms of worst-case bounded error.

### 4.4 Comparison of Numerical Formats

Based on the derivations in Theorems 1 through 5, we observe a distinct hierarchy of resilience to bit-flip errors. The fundamental difference lies in how the bit representation maps to the value: integer errors are *additive* and linear with respect to bit position, whereas floating-point errors are *multiplicative* and exponential with respect to bit position (specifically the exponent).

### 4.4.1 Hierarchy of Resilience for DNNs

**1. BNN (Very High Resiliency)**
The extreme case of Integer representation using only 1 bit, the sign bit. In this special case, a bit flip contributes to a small constant error.

**2. Integer / Fixed-Point (High Resiliency)**
A bit flip contributes an additive error proportional to $2^k$. The maximum possible error is bounded by the dynamic range of the integer, which scales as $2^B$. The variance of the error is constant across all weights of the same bit-width, regardless of the weight's value.

**3. Affine Quantization (AQ) (Medium Resiliency)**
This introduces a shared-failure mode absent in pure integer representations. While the bulk of the parameters (the weights) are robust integers, corruption of the shared zero-point $Z$ or scale $S$ affects the entire layer simultaneously. A corruption in $Z$ introduces a correlated layer-wide term proportional to $\left(\sum_i x_i\right)^2 \mathrm{Var}(Z')$, while corruption in the floating-point scale $S$ multiplicatively distorts the full accumulation. As a result, layer-wise affine quantization is generally less resilient than pure integer schemes, although still more resilient than standard floating-point weights in expectation.

**4. Floating-Point (Low Resiliency)**
Floating-point is the most fragile representation. Errors in the exponent bits are multiplicative. If the MSB of the exponent flips, the value can change by several orders of magnitude (e.g., $10^{-3} \to 10^4$). The exponent contribution can induce very large multiplicative distortions. In the worst case, the second moment grows doubly-exponentially with the exponent width, consistent with the upper bound derived in the floating-point analysis. Consequently, a single bit flip in a standard FP32 weight can result in an error magnitude that dwarfs the signal of the entire network.

### 4.4.2 Summary of Error Scaling

Table 1 summarizes the error characteristics. Note the difference in scaling between Integer (exponential in $B$) and Floating-Point (double exponential in $E$).

Table 1: Comparison of Bit-Flip Resilience by Numerical Format

| Format | Error Type | Dominant Noise Source | Max Error Scaling | Resilience |
|---|---|---|---|---|
| **BNN** | Additive | Sign | Constant | **Very High** |
| **Integer** | Additive | Magnitude Bits | $O(2^B)$ | **High** |
| **AQ** | Additive/Multiplicative | Zero-Point / Scale | $O(2^{2^E})$ (via Scale) | **Medium** |
| **Float (FP)** | Multiplicative | Exponent Bits | $O(2^{2^E})$ | **Low** |

## 4.5 Error Propagation through Activation Functions

We analyze the expected squared error (MSE) at the output of the activation function $a = \phi(y)$. Let $y$ be the clean pre-activation and $y' = y + \xi$ be the corrupted pre-activation, where $\xi$ represents the total weight error noise. We do not assume $\xi$ is small, as bit flips can induce large perturbations.

### 4.5.1 Rectified Linear Unit (ReLU)

Let $\phi(x) = \max(0, x)$. The error depends on the regime of the clean input $y$ relative to the noise $\xi$.

**Theorem 6** (Expected MSE for ReLU). *Given clean input $y$ and noise $\xi$ with PDF $f_\xi(\xi)$, the expected MSE $E[(\Delta a)^2]$ is:*

$$E[(\Delta a)^2] = \int_{-y}^{\infty} \xi^2 f_\xi(\xi) d\xi + \int_{-\infty}^{-y} (-y)^2 f_\xi(\xi) d\xi \quad (\text{for } y > 0) \tag{35}$$

*Proof.* Let $\Delta a = \text{ReLU}(y+\xi) - \text{ReLU}(y)$. Case 1: $y > 0$. If $\xi > -y$, then $y+\xi > 0$, so $\Delta a = (y+\xi) - y = \xi$. If $\xi \leq -y$, then $y+\xi \leq 0$, so $\Delta a = 0 - y = -y$. Taking the expectation over $\xi$: $E[\Delta a^2] = \int_{-y}^{\infty} \xi^2 f(\xi)d\xi + \int_{-\infty}^{-y} y^2 f(\xi)d\xi$. (The case for $y \leq 0$ is symmetric). $\qquad\square$

**Implication:** ReLU offers **no error attenuation** for positive perturbations. If $\xi$ is large and positive, $\Delta a = \xi$. The error propagates linearly and unbounded.

### 4.5.2 Sigmoidal Activations with Temperature

Consider $\sigma_\tau(x) = \frac{1}{1+e^{-x/\tau}}$. We analyze resilience to large noise $\xi$ (e.g., bit flips).

**Theorem 7** (Sigmoid Saturation Resilience)**.** *Let $y$ be a clean input such that $|y| \gg \tau$ (saturated regime). Let $\xi$ be a noise term. As $\tau \to 0$ (Step Function limit), the expected MSE converges to the probability of a sign flip:*

$$\lim_{\tau \to 0} E[(\sigma_\tau(y+\xi) - \sigma_\tau(y))^2] = P(sign(y+\xi) \neq sign(y)) \tag{36}$$

*Proof.* Let $\tau \to 0$. Then $\sigma_\tau(x) \to \mathbb{I}(x > 0)$. The error is $(\mathbb{I}(y+\xi > 0) - \mathbb{I}(y > 0))^2$. Since $\mathbb{I} \in \{0,1\}$, the difference is non-zero (value 1) if and only if the signs differ. Thus, $E[\Delta a^2] = 1 \cdot P(\text{Sign Mismatch})$. $\qquad\square$

**Comparison with High Temperature ($\tau \to \infty$):** For high $\tau$, the sigmoid is linear $\sigma_\tau(x) \approx \frac{1}{2} + \frac{x}{4\tau}$. The error is:

$$E[\Delta a^2] \approx \frac{1}{16\tau^2} E[\xi^2] \tag{37}$$

While this attenuates the variance by $1/\tau^2$, it scales with the magnitude of the noise $E[\xi^2]$.

### 4.5.3 Comparison of Activation Resilience

Table 2 summarizes the resilience. We conclude that for hardware faults where $E[\xi^2]$ is potentially unbounded (e.g., exponent flips), **Bounded Activations (Low $\tau$)** are strictly superior to Unbounded Activations (ReLU/High $\tau$).

Table 2: Resilience to Large-Magnitude Bit Errors

| Function | Mechanism | Error Scaling | Rank |
|---|---|---|---|
| **Step / Low-$\tau$** | Saturation (Masking) | $P(\text{Sign Flip})$ (Bounded) | **1** |
| **Tanh / Sigmoid** | Saturation (Damping) | Bounded Constant | **2** |
| **High-$\tau$ Sigmoid** | Linear Attenuation | $\propto \text{Var}(\xi)$ (Unbounded) | **3** |
| **ReLU** | Pass-through | $\propto \text{Var}(\xi)$ (Unbounded) | **4** |

## 4.6 Width and Sparsity

We analyze how the total output error scales with the fan-in $n$ (width) and the input sparsity $k$ (number of non-zero elements in $x$).

### 4.6.1 Effect of Width ($n$)

Assume inputs $x_i$ are drawn from a distribution with variance $\sigma_x^2$.

- **Variance Term (Incoherent Error):** Scales linearly with width.

$$\text{Var}(y') \propto \sum_{i=1}^{n} x_i^2 \approx n\sigma_x^2 \tag{38}$$

- **Bias Term (Coherent Error):** Scales quadratically with width.

$$\text{Bias}(y')^2 \propto \left(\sum_{i=1}^{n} x_i\right)^2 \approx n^2 \mu_x^2 \tag{39}$$

**Analysis:** If the weight errors have a non-zero mean ($E[\Delta w] \neq 0$, which is true for all discussed formats due to sign-bit asymmetry or bias), wider networks suffer disproportionately from **Bias Accumulation**. While the signal grows as $n$, the bias error grows as $n^2$, causing the Signal-to-Noise Ratio (SNR) to degrade in very wide, dense layers unless specific compensation is applied.

### 4.6.2 Effect of Sparsity

Let $k$ be the number of non-zero elements in $x$ (L0 norm). The error terms depend only on active inputs.

$$E[(y' - y)^2] \propto k \cdot \text{Var}(\Delta w) + k^2 \cdot E[\Delta w]^2 \tag{40}$$

**Implication:** High activation sparsity (e.g., from ReLU) acts as a natural error suppressor. When only a small fraction of inputs are active, both the variance and bias terms are reduced relative to a dense layer, with the leading-order variance contribution scaling linearly in the number of active inputs.

### 4.7 Depth

We model the error propagation across a network of depth $L$. Let $e_l^2 = E[(y_l' - y_l)^2]$ be the MSE at layer $l$. Assume a simplified layer response with Lipschitz constant (gain) $\lambda$.

$$e_l^2 \approx \lambda^2 e_{l-1}^2 + \text{Noise}_l \tag{41}$$

where $\text{Noise}_l$ is the intrinsic error generated by bit flips in layer $l$ itself.

Under a simplified error-propagation model in which each layer contracts or amplifies the previous-layer MSE by a uniform factor $\lambda^2$ and contributes additive intrinsic noise $\nu$, the end-to-end MSE satisfies the following recursion.

**Theorem 8** (Depth Accumulation under a Uniform Gain Model)**.** *For a network of depth $L$ with uniform layer gain $\lambda$ and intrinsic noise variance $\nu$:*

$$e_L^2 = \nu \sum_{k=0}^{L-1} (\lambda^2)^k = \nu \frac{1 - (\lambda^2)^L}{1 - \lambda^2} \tag{42}$$

**Regimes of Stability:**

- **Attenuating ($\lambda < 1$):** The error stabilizes to a finite asymptote $\frac{\nu}{1-\lambda^2}$. The network is robust to infinite depth.

- **Exploding ($\lambda > 1$):** The error grows exponentially with depth $O(\lambda^{2L})$. Deep networks without residual connections or careful normalization are highly susceptible to "error avalanches."

- **Critical ($\lambda = 1$):** The error grows linearly $O(L)$.

### 4.8 The Path to Weightless Networks

The preceding analysis identifies the structural ingredients that improve resilience under the bit-flip model studied in this paper. Taken together, these results point directly toward sparse, discrete, logic-based computation. In particular, they isolate three recurring ingredients of resilient architectures:

1. **Minimal Precision:** Theorems 1 and 2 demonstrate that error variance scales with the dynamic range of the weights. Integer and binary representations ($B = 1$) minimize this range, eliminating the "explosive" error modes of floating-point exponents.

2. **Hard Saturation:** The analysis of activation functions shows that "step-like" functions (low temperature $\tau$) provide the optimal masking capability for large-magnitude faults, bounding the error to a simple sign flip rather than permitting unbounded noise propagation.

3. **High Sparsity:** The width-scaling analysis reveals that bias accumulation grows quadratically with fan-in ($n^2$). Reducing fan-in (sparsity) is essential to check this growth and localize the impact of faults.

### 4.8.1 From BNN to LUT-Based Models

Taken together, these trends define a clear architectural progression away from dense, high-precision weighted networks and toward sparse, discrete, logic-based systems.

**Step 1: Sparse Binary Neural Networks (BNNs)**
Combining minimal precision (1-bit weights) with hard saturation (Sign activation) yields the Binary Neural Network. While robust, standard BNNs typically remain dense (high fan-in), leaving them vulnerable to error propagation across the fully connected structure.

**Step 2: LogicNets (Fixed-Function Sparse BNNs)**
By enforcing extreme sparsity on a BNN (e.g., fan-in $K \leq 6$), each neuron's behavior becomes deterministic over a small, enumerable input space. Such a neuron is mathematically equivalent to a fixed Boolean function and can be "compiled" into a Lookup Table (LUT) for inference (Umuroglu et al., 2020). This structural shift effectively discretizes the error landscape: a fault no longer affects a weight (which multiplies all inputs) but flips a single entry in a truth table, isolating the error to a specific input pattern.

**Step 3: Advanced LUT Mappings (PolyLUT, NeuraLUT)**
While LogicNets compile trained BNNs into LUTs, subsequent approaches like PolyLUT (Andronic & Constantinides, 2023) and NeuraLUT (Andronic & Constantinides, 2024) seek to increase expressivity by mapping more complex functions (e.g., neurons with skip connections or polynomial approximations) into the same LUT fabric. These methods improve performance but still rely on an underlying weight-based abstraction during training.

**Step 4: Differentiable Weightless Neural Networks (DWNs)**
The final evolutionary step is to abandon the weight proxy entirely. DWNs (Bacellar et al., 2024b) learn the LUT contents directly via gradient descent (using relaxations like the Extended Finite Difference method). By treating the LUT entry itself as the learnable parameter, DWNs align the training objective directly with the resilient substrate. This architecture represents the convergence of our theoretical ideals: strictly bounded parameters (bits), high sparsity (localized LUTs), and discrete step-function activations (table lookups).

## 4.9 Theoretical Analysis of LUT-Based Resilience

In this section, we formalize the resilience properties of Lookup Table (LUT)-based networks. We prove a bounded error-isolation property for LUT neurons and then analyze a novel "Symmetric Recovery" phenomenon that arises under extreme corruption.

### 4.9.1 Definitions

Let a LUT layer consist of neurons where each is a $K$-input LUT implementing a Boolean function $f : \{0,1\}^K \rightarrow \{0,1\}$. The LUT is stored as a vector $T \in \{0,1\}^N$ where $N = 2^K$. For an input address $x \in \{0,1\}^K$, the output is $y = T[x]$. We define the Bit Error Rate (BER) $p$ as the probability that any storage bit (in $T$ or the input $x$) is flipped.

### 4.9.2 Bounded Error Propagation

**Theorem 9** (LUT Error Isolation)**.** *In a LUT-based neuron with $N = 2^K$ entries, a single bit flip in the parameter memory $T$ affects the output for exactly one input pattern out of $2^K$ possible patterns. The expected absolute error for a uniformly distributed input is minimized as $K$ increases.*

*Proof.* Let $T'$ be the corrupted table with a single bit flip at index $j$. For an input $x$, the output error is $\Delta y = |T'[x] - T[x]|$. Since only the $j$-th bit is flipped, $\Delta y = 1$ if $x = j$, and 0 otherwise. Assuming inputs $x$ are uniform over $\{0, \ldots, N-1\}$, the expected error is:

$$E[|\Delta y|] = \sum_x P(x)\Delta y(x) = \frac{1}{N} \cdot 1 = \frac{1}{2^K} \tag{43}$$

In contrast, a weight bit flip in a standard neuron affects the dot product for *all* inputs where the corresponding input feature is non-zero. Thus, LUTs provide an exponential reduction in error probability per fault. $\square$

### 4.9.3 Symmetry and Recovery at High Corruption

We observe a counter-intuitive regime in which accuracy can partially recover as the corruption rate approaches $p = 1$. This phenomenon is tied to a structural tendency often observed in trained LUTs: complementary addresses $\overline{x}$ and $x$ frequently store opposite values. Intuitively, complementary binary patterns often correspond to semantically opposing configurations, so a LUT that activates on $x$ may tend to deactivate on $\overline{x}$, and vice versa.

We formalize this tendency through an anti-symmetry probability. For a LUT $T : \{0,1\}^K \to \{0,1\}$, define

$$\alpha := P(T[\overline{x}] \neq T[x]),$$

where the probability is taken over the input distribution on addresses $x$. The quantity $\alpha$ measures how often complementary addresses store opposite values. When $\alpha$ is large, the LUT is more anti-symmetric; when $\alpha = 1$, the LUT is perfectly anti-symmetric.

**Mechanism of Cancellation.** Consider a fully corrupted state ($p = 1$). For a LUT neuron with input address $x$:

1. **Input Inversion:** the input address $x$ becomes its bitwise complement $\overline{x}$;

2. **Content Inversion:** the stored LUT contents $T$ become their bitwise complement $\overline{T}$.

The corrupted output is therefore $y' = \overline{T}[\overline{x}]$. Since $\overline{T}[k] = \neg T[k]$, this becomes $y' = \neg T[\overline{x}]$, while the clean output is $y = T[x]$. Hence, exact recovery occurs if and only if

$$y' = y \iff \neg T[\overline{x}] = T[x] \iff T[\overline{x}] \neq T[x].$$

Thus, recovery under full corruption is governed exactly by anti-symmetry across complementary addresses.

**Theorem 10** (Conditional Symmetric Recovery at Full Corruption)**.** *Consider a LUT neuron under full corruption $p = 1$. If $\alpha := P(T[\overline{x}] \neq T[x])$, then the probability of exact recovery is exactly*

$$P(y' = y) = \alpha.$$

*Proof.* Under full corruption, the address is inverted and the LUT contents are complemented, so $y' = \overline{T}[\overline{x}] = \neg T[\overline{x}]$. The clean output is $y = T[x]$. Therefore, $y' = y$ if and only if $T[\overline{x}] \neq T[x]$. Taking probability over the input distribution yields $P(y' = y) = P(T[\overline{x}] \neq T[x]) = \alpha$. $\square$

This theorem is exact and conditional. The empirical or structural question is not whether recovery follows from anti-symmetry—it does—but whether trained logic/LUT networks tend to realize large values of $\alpha$. Our experiments suggest that they often do strongly enough for the recovery effect to become visible at the network level.

**Depth-Dependent Recovery.** The theorem above characterizes recovery for a single LUT neuron. We now extend the argument to deep compositions.

**Corollary 1** (Guaranteed Even-Layer Recovery in the Perfect Anti-Symmetric Case)**.** *Consider a logic/LUT network of even depth $L$ under full corruption $p = 1$. If every LUT along the active computation path satisfies $T[\overline{x}] \neq T[x]$ for all admissible addresses $x$ (equivalently, $\alpha = 1$ at every layer), then the network recovers the clean computation exactly.*

*Proof.* By the theorem, when $\alpha = 1$, each corrupted LUT reproduces its clean output exactly under $p = 1$. Therefore, each layer outputs exactly its clean activation, and this recovery propagates through the entire composition. Hence the final network output is recovered exactly. □

This depth dependence is not exact in general, since recovery events across layers need not be independent. The following scaling should therefore be interpreted as a simplifying approximation that captures the expected weakening of recovery with depth.

When $\alpha < 1$, exact cancellation is no longer guaranteed at every layer. Instead, recovery at each layer or cancellation block succeeds only with probability $\alpha$, so the end-to-end recovery probability decreases multiplicatively with depth. Under a simplifying independence approximation across layers, a depth-$L$ even-layer network satisfies $P(\text{exact recovery}) \approx \alpha^L$. If recovery is analyzed at the level of cancellation pairs, this can equivalently be written as $P(\text{exact recovery}) \approx \alpha^{L/2}$, where $\alpha$ is then interpreted as the per-pair recovery probability. In either case, whenever $\alpha < 1$, the product of multiple factors smaller than one decreases monotonically with depth. Thus, deeper even-layer networks still exhibit the same recovery mechanism, but the strength of the recovery weakens as depth increases.

**Corollary 2** (Depth Weakens Even-Layer Recovery)**.** *At full corruption ($p = 1$), even-layer networks can exhibit recovery through the symmetric cancellation mechanism, but the strength of this recovery decreases with depth whenever $\alpha < 1$. Consequently, shallow even-depth networks recover more strongly than deeper even-depth networks.*

## 5 Experimental Evaluation

To validate the theoretical analysis derived in Section 4 and confirm the hypothesis that resilience is a structural property converging towards logic-based architectures, we conducted extensive fault injection simulations. The experiments were designed to isolate specific architectural hyperparameters—numerical precision, width, depth, activation functions, and sparsity—to quantify their individual contributions to bit-flip resilience.

### 5.1 Experimental Setup

**Datasets and Benchmarks:** We evaluated resilience across the **MLPerf Tiny** benchmark suite (Banbury et al., 2021), which represents the standard for resource-constrained edge inference. The tasks include Image Classification (MNIST (Deng, 2012), Fashion-MNIST (Xiao et al., 2017)), Keyword Spotting (Google Speech Commands), and Anomaly Detection (ToyAdmos).

**Models:** For the architectural ablations, whose goal is to isolate the effect of individual hyperparameters, we use Multi-Layer Perceptrons (MLPs). Unless otherwise stated, the ablation baseline is an FP32 one-hidden-layer MLP with 256 hidden neurons and ReLU activation, and each ablation varies only the factor under study: precision, width, depth, activation, or sparsity. This choice is primarily driven by computational cost: our evaluation requires dense sweeps over corruption rates together with Monte Carlo fault injection (approximately 100 trials per point), and repeating the full study with larger CNN baselines across every ablation would be prohibitively expensive. The MLP setting is sufficient to expose the structural scaling trends of precision, width, depth, activation, and sparsity in a controlled and computationally tractable form. For the final Logic/LUT comparison, specifically against the Differentiable Weightless Network (DWN) (Bacellar et al., 2024b) under high-corruption settings, we additionally report results against the standard

MLPerf Tiny CNN baselines to show that the same resilience trends observed in the controlled MLP study also appear in representative edge architectures.

**Fault Injection Protocol:** We model faults as independent Bernoulli trials where each bit in the binary representation of the network parameters (weights, LUTs) flips with probability $p$. We evaluate a comprehensive spectrum of Bit Error Rates (BER), from "benign" hardware faults ($p \in [10^{-8}, 10^{-3}]$) to "catastrophic" failure modes ($p \in [0.01, 1.0]$).

## 5.2 The Hierarchy of Precision

We first validate the impact of numerical precision on resilience. Figure 2 illustrates the degradation of accuracy as the bit error rate increases across FP32, FP16, FP8, INT8, INT4, INT2, and BNN formats. For clarity, all INT results reported here use *layer-wise affine quantization*, not pure integer quantization. Pure integer quantization was not included because, while it is theoretically more resilient to bit flips, it did not provide sufficient baseline task accuracy in our experiments. Thus, the INT curves shown here reflect the practically relevant quantized setting used in deployment.

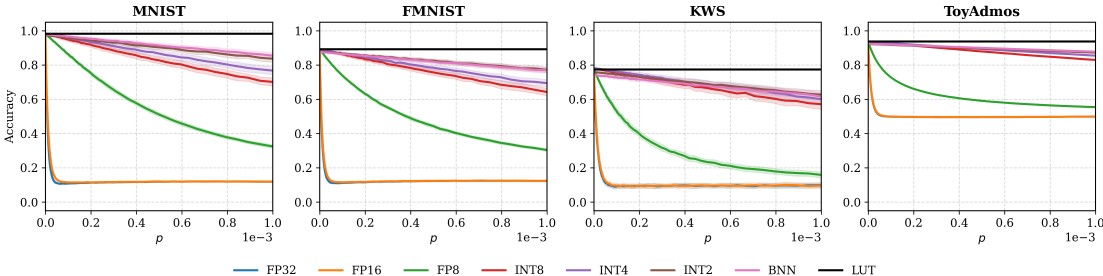

Figure 2: **Numerical Precision Ablation.** Accuracy vs. Bit Error Rate ($p$) averaged across datasets. A clear hierarchy emerges: floating-point models collapse early ($p \approx 10^{-5}$) due to exponent errors, while integer and binary models sustain accuracy significantly longer.

The results confirm a strict hierarchy where **lower precision yields higher resilience**:

- **Floating Point (FP32/16):** These formats exhibit a "cliff-edge" failure mode, dropping to random accuracy at relatively low error rates ($p \approx 10^{-5}$). This confirms Theorem 2, where exponent bit flips cause unbounded multiplicative errors.

- **Binary (BNN):** 1-bit weights proved significantly more robust, maintaining usable accuracy up to $p \approx 10^{-3}$. By eliminating the exponent and magnitude bits entirely, BNNs bound the maximum error per weight.

- **Logic/LUT:** Most notably, when comparing the weighted baselines to LUT-based models (DWNs), the gap is substantial. In the $10^{-3}$ regime, where FP32 and FP16 models drop roughly 62% accuracy on average, LUT-based models show $\approx 0$ degradation. Higher fault rates (up to 40%) are discussed in section 5.4.2.

These results also illustrate the accuracy–resilience trade-off induced by representation choice. Models with higher clean accuracy at $p = 0$ can degrade much faster under memory corruption, while lower-precision and logic/LUT-based models may sacrifice a small amount of clean accuracy but preserve performance over a wider BER range. This trade-off is visible, for example, in KWS, where FP32 begins with slightly higher clean accuracy than BNN but collapses much earlier as the corruption rate increases.

## 5.3 Architectural Ablations

Having established that minimal precision (and, in the binary limit, 1-bit) is strongly favored under the corruption model, we analyze the remaining structural components—Width, Depth, Activation, and Sparsity—

to demonstrate that resilience improves as these parameters approach the configuration of a Logic/LUT network.

### 5.3.1 Width

We varied the width of the hidden layers in the MLP baseline from $W = 64$ to $W = 1024$ while keeping depth constant.

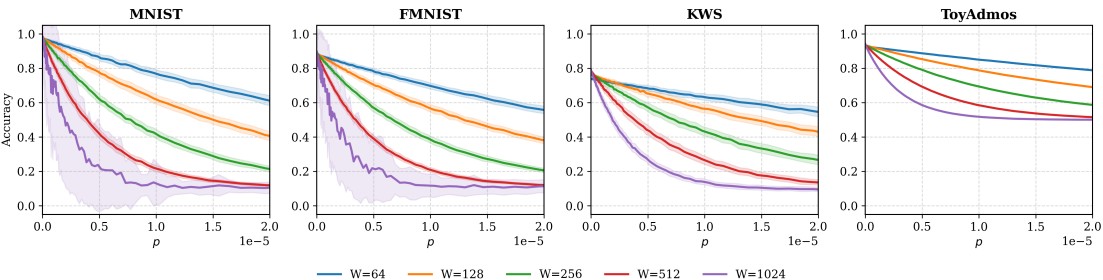

Figure 3: **Width Ablation.** Wider networks (e.g., $W = 1024$) degrade faster than narrower ones, validating the $O(n^2)$ bias accumulation hypothesis.

As shown in Figure 3, wider networks degrade significantly faster than narrower ones. For example, at $p = 10^{-5}$ on MNIST, the $W = 1024$ model suffers a nearly 40% accuracy drop, whereas the $W = 64$ model remains robust. This empirically validates the bias accumulation term derived in Equation 18 ($Bias^2 \propto n^2$). Resilience favors narrow fan-in, which logic-based networks enforce by design (typically $N \leq 6$).

### 5.3.2 Depth

We analyzed error propagation in networks of increasing depth ($L = 1$ to $6$).

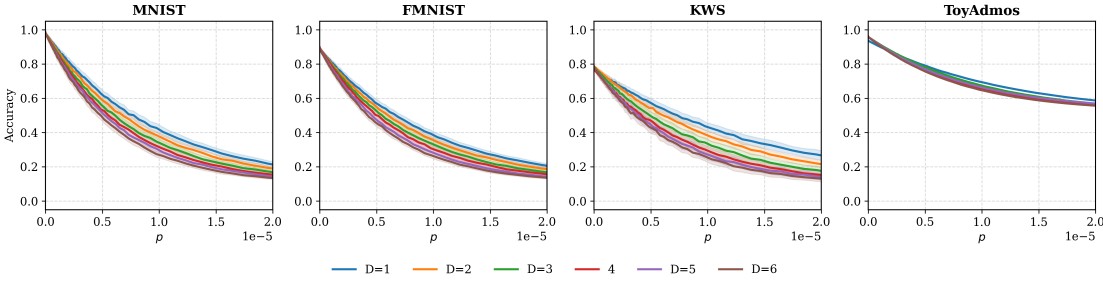

Figure 4: **Depth Ablation.** Deeper networks suffer from error avalanches, confirming the multiplicative error propagation in standard architectures.

We observe a consistent decrease in robustness as depth increases (Figure 4). The "error avalanche" effect is visible, where noise introduced in early layers is amplified by subsequent transformations. This confirms that shallow networks are more robust, aligning with DWN architectures which typically utilize shallow lookup layers.

### 5.3.3 Activation Function

We compared the resilience of unbounded activations (ReLU) against bounded activations (Sigmoid, Tanh) and hard-saturation functions (Sign/Step).

Figure 5 provides strong evidence for the hard saturation hypothesis. ReLU (unbounded) fails earliest. Conversely, Sign/Step functions offer the highest resilience. By forcing saturation, these functions mask

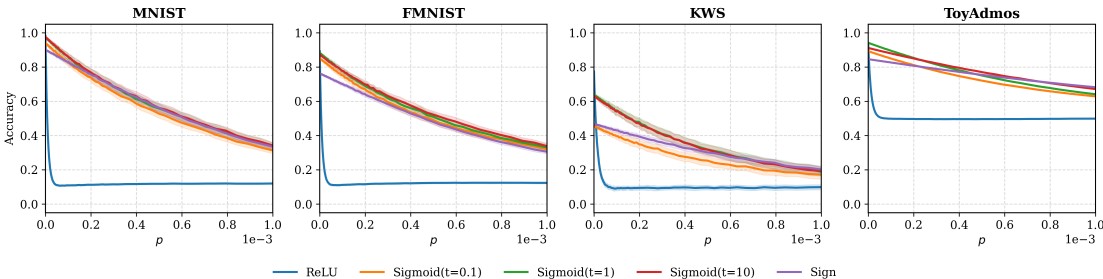

Figure 5: **Activation Function Ablation.** Hard saturation functions (Sign, low-temp Sigmoid) provide superior error masking compared to unbounded ReLU.

small-to-moderate perturbations entirely and convert large perturbations into binary bit-flips. The most resilient activation is the Step function—the native behavior of a LUT.

### 5.3.4 Sparsity

We evaluated unstructured sparsity levels ranging from 0% (dense) to 99% (highly sparse) using an "Iso-Model Size" comparison to ensure fair bit-budgeting.

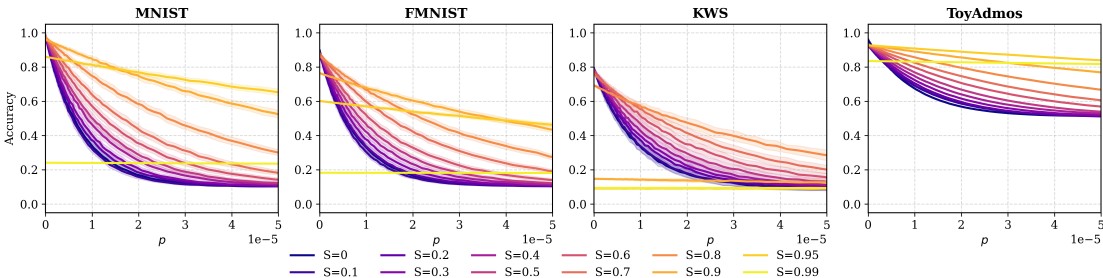

Figure 6: **Sparsity Ablation.** Extreme sparsity ($> 90\%$) acts as a dominant resilience factor, significantly delaying accuracy collapse.

Figure 6 demonstrates that sparsity is a dominant factor in resilience. Even after accounting for the storage overhead, highly sparse models ($S \geq 90\%$) consistently outperform dense models. This aligns with Eq. 19, where the total error variance scales linearly with the number of active elements. Logic-based networks are intrinsically sparse, maximizing this benefit.

### 5.4 Logic/LUT Resiliency

We next compare the Differentiable Weightless Network (DWN) against the weighted baselines. Relative to the preceding ablations, Logic/LUT-based models combine low-precision discrete parameters, bounded Boolean outputs, localized connectivity, and shallow compositions. In addition, as shown in the LUT analysis, a bit flip in a table entry produces a bounded error that affects only the corresponding addressed outputs, rather than an arithmetic perturbation that propagates through a continuous weight value.

### 5.4.1 Small-Range Corruption Immunity

In the standard operating regime of hardware faults ($p \leq 10^{-4}$), the performance gap between arithmetic and logic-based models is significant. As shown in Figure 2, the Logic/LUT-based models remain essentially unchanged in this regime, whereas the weighted baselines begin to degrade. Across our Monte Carlo trials, we observe no measurable average accuracy drop for the DWN models at these corruption rates.

### 5.4.2   Catastrophic Fault Regime ($p = 0.01$ to $0.4$)

The structural superiority of weightless logic becomes most apparent when subjected to catastrophic bit error rates that could result from faults or bit-flip attacks. As illustrated in Figure 7, the FP32, FP16, FP8, and BNN results in this comparison use the standard MLPerf Tiny CNN baselines, while the Logic/LUT curve uses the DWN model. The standard numerical CNN baselines suffer a total collapse to random-model accuracy at the first injection step of $p = 0.01$, with the BNN baseline following shortly after.

In contrast, the Logic/LUT-based architecture maintains functional utility far beyond its weighted counterparts:

- **At 0.1 corruption:** While weighted models have already reached random-guess levels (yielding approximately -65% drop), LUT-based NNs exhibit a marginal 2% drop in accuracy.

- **At 0.2 corruption:** Even at this severe fault rate, where BNNs have completely failed, Logic/LUT models sustain high performance with only an 8% drop.

- **The 0.5 Threshold:** The weightless model only reaches the mathematical limit of random guessing at 50% corruption. This is the theoretical point where a model is statistically indistinguishable from a randomly initialized one, yet the DWN remains the only architecture capable of approaching this limit gracefully.

These results are consistent with the theoretical picture developed in Section 4: shifting from continuous weight-based multiplication to discrete Boolean lookups replaces rare, potentially large multiplicative distortions with a bounded address-localized failure mechanism.

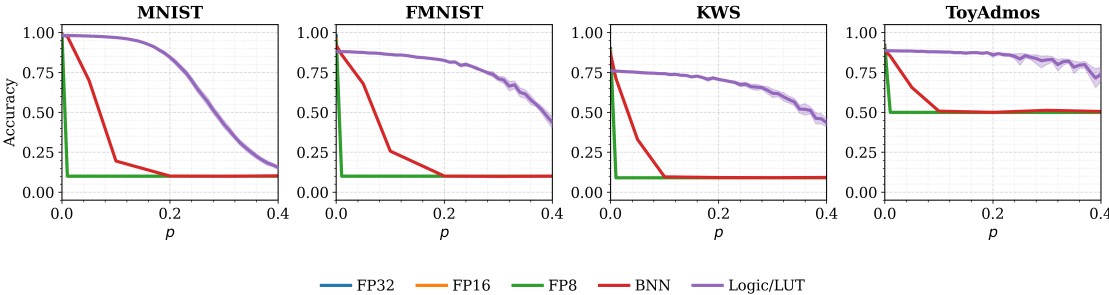

Figure 7: Parameter bit-flip resilience on MNIST, FashionMNIST, KWS and ToyAdmos datasets with corruption rates ranging from 0 to 40%. FP32 and FP16 lines are overlapped with FP8.

### 5.5   Symmetric Recovery in Logic/LUT Networks

Finally, we investigate the unique "Symmetric Recovery" phenomenon predicted for Logic-Based architectures. We trained DWN models of varying depths and subjected them to the full range of bit error rates up to $p = 1.0$.

Figure 8 reveals a striking behavior as the corruption rate approaches 1.0. Networks with an even number of layers ($L = 2, 4, 6$) recover substantially from the random-guess regime, consistent with the cancellation mechanism $y' = \neg T[\neg x]$. In contrast, odd-depth networks tend toward inversion rather than recovery. We emphasize that the exact identity is algebraic, while the magnitude of the observed recovery depends on how strongly the trained LUTs exhibit anti-symmetry across complementary addresses.

## 6   Conclusion

In this work, we studied resilience to parameter bit flips as a structural property of neural computation. Through neuron- and layer-level analysis, we derived expected error expressions for multiple numerical

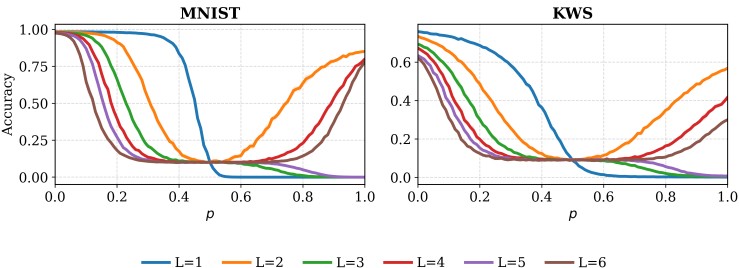

Figure 8: **DWN Layer Parity Recovery.** for corruption rates $p$ ranging from 0 to 100% and network depths $L = 1$ to 6. As $p \rightarrow 1$, even-depth networks exhibit partial recovery consistent with symmetric error cancellation across layers, whereas odd-depth networks tend toward inverted outputs.

formats and architectural primitives, and used them to identify a consistent set of resilience-favoring trends: lower precision, higher sparsity, bounded activations, and shallower compositions. We then argued that Logic and Lookup-Based Neural Networks instantiate an extreme point in this design space.

Our experiments on the MLPerf Tiny benchmark suite are consistent with this picture. In particular, Logic/LUT-based architectures such as Differentiable Weightless Networks (DWNs) remain highly stable in corruption regimes where conventional floating-point models degrade sharply. In addition, the discrete structure of LUT-based networks gives rise to a distinctive even-layer recovery effect at extreme corruption rates, which we analyzed through the interaction between address inversion and table inversion.

Taken together, these results suggest that moving from continuous arithmetic weights to discrete Boolean lookup mechanisms can offer an attractive accuracy–resilience trade-off for fault-tolerant edge inference. More broadly, the paper points toward a design principle for reliable neural systems: resilience can be shaped directly at the representational and architectural level, rather than treated only as a property of training or hardware protection. Future work includes extending these principles to larger architectures, tightening the theory of recovery effects in trained LUTs, and evaluating structured hardware fault models beyond independent bit flips.

## Acknowledgements

This research was supported by Semiconductor Research Corporation (SRC) Task 3148.001, National Science Foundation (NSF) Grants #2326894, #2425655 (supported in part by the federal agency and Intel, Micron, Samsung, and Ericsson through the FuSe2 program), NVIDIA Applied Research Accelerator Program, and compute resources on the Vista GPU cluster through CGAI & TACC at UT Austin. Any opinions, findings, conclusions, or recommendations are those of the authors and not of the funding agencies.

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

# A  Training Runtime and Model Size

Table 3 reports average training runtime and storage size for the main model variants on the A100 NVIDIA GPU. The model-size column counts the bits required to store the parameters used during inference; for affine quantized models, this includes the integer and the stored scale/zero-point metadata.

Table 3: Training runtime and model size for different model variants.

| Model | Avg. Training time | MNIST | FMNIST | KWS | ToyAdmos |
|---|---|---|---|---|---|
| FP32 | 8 min | 795 KiB | 795 KiB | 14 MiB | 14 MiB |
| FP16 | 8 min | 397 KiB | 397 KiB | 7 MiB | 7 MiB |
| FP8 | 8 min | 199 KiB | 199 KiB | 3 MiB | 3 MiB |
| INT8 | 8 min | 199 KiB | 199 KiB | 3 MiB | 3 MiB |
| INT4 | 8 min | 99 KiB | 99 KiB | 1.8 MiB | 1.8 MiB |
| INT2 | 8 min | 50 KiB | 50 KiB | 0.9 MiB | 0.9 MiB |
| BNN | 10 min | 25 KiB | 25 KiB | 0.4 MiB | 0.4 MiB |
| Logic/LUT | 10 min | 6 KiB | 8 KiB | 0.003 MiB | 0.003 MiB |

