# OpenReview forum: "From Arithmetic to Logic: The Resilience of Logic and Lookup-Based Neural Networks Under Parameter Bit-Flips"
_TMLR — Accepted by TMLR_

### Review · Reviewer_9EEf · 2026-03-29

**Summary Of Contributions:**

The paper studies the reslience of different forms of neural networks under bit flip errors in the network's parameter values (caused by hardware failure, for instance). In particular, the paper studies lookup table networks (LUTs), a form of network in which each layer essentially maps possible binary inputs to different binary outputs in a table. The paper first shows theoretically that while typical network formats like floating point, resilience is low due to the large effect of corrupting exponent bits. On the other hand, for more quantized networks (in particular, binary neural networks and LUTs), resilience is very high. The authors also study the resilience of other architectural features like activations, width and depth. The theory is then validaded on several benchmark datasets.

**Audience:**

Yes

**Audience Explanation:**

Overall, the results seem pretty relevant to the community studying bit-level parameter robustness. From an infrastructure perspective, this topic seems relevant in settings where hardware-level resilience is not an option.

However, I'm not sure about the practical significance of the symmetry results under a p=1 corruption rate. In my view, any corruption rate even approaching p=0.5 is practically highly unlikely (since this is essentially randomizing all bits), so going beyond p=0.5 seems practically irrelevant (although the symmetry results are interesting).

**Broader Impact Concerns:**

No broader impact concerns.

**Claims And Evidence:**

Yes

**Claims Explanation:**

Largely speaking, the theory seems solid. Proofs appear correct and the big-picture conclusion that LUTs are more robust to bit-flips is both intuitively reasonable and well-backed up.

There are a couple of important concerns I have though. First, the authors mostly consider resilience at the level of individual layers or features of an architecture (e.g. width, depth or single weight layers). It would also be useful to have some kind of resilience guarantee on an entire network- the width and depth results somewhat approach this, but rely on rough heuristics.

The second concern is experimental: some important details seems to be missing regarding the experimental setup. In particular, how are different networks chosen so that they all achieve the same/similar performance at p = 0 (this is relevant to all the results figures)? Another practically relevant question is computational cost: how long does it take to train an LUT compared to a floating point network to the same performance level? Also, the memory size of different networks doesn't seem to be mentioned. Practitioners may want to know what the relative advantages of LUTs vs other networks are overall (beyond just robustness to bit-flips).

**Requested Changes:**

**Critical**
- Please provide resilience guarantees at a network-wide level, or comment on how layer-wise guarantees can be composed to produce a full-network guarantee
- Missign experimental detail: how are different networks trained so that they all have the same performance at p=0?
- Please include runtimes to train networks of different precision levels, as well as including the size of different networks (in terms of number of bits)

**Would strengthen**
- Text in some figures is a bit small

---

> ### Author Response · Authors · 2026-05-17
>
> We thank the reviewer for the positive assessment and constructive suggestions.
>
> **Network-wide resilience guarantees:** An exact resilience guarantee for an entire arbitrary network is exponentially hard and highly dependent on the specific network topology, connectivity pattern, nonlinearities, and parameter representation. Our goal is therefore to provide what is both tractable and practically useful: a component-wise structural analysis showing how each architectural choice affects the expected corruption-induced error, and how these effects are expected to compose as the network changes in width, depth, precision, activation function, and sparsity.
>
> In particular, the paper derives expected error expressions for individual numerical formats and layer primitives, and then analyzes how these errors propagate through depth using the depth-composition model. Thus, while the layer-wise results are local, they are not isolated: the depth analysis provides the mechanism by which layer-level errors compose across the full network. We have added text in the revised version to make this connection clearer.
>
> **Training and clean performance at p = 0.:**
> The different networks were not constrained to have exactly identical performance at p=0. In the ablation studies, we keep the underlying setup fixed and vary only the hyperparameter under study, such as precision, width, depth, activation, or sparsity. As a result, there are clean-accuracy differences at p=0, but these differences are generally small relative to the degradation observed as the bit error rate increases. This can make the curves appear visually similar at p=0. For example, in KWS, the FP32 model starts at approximately 78% accuracy, while the BNN starts at approximately 74%; however, at higher corruption rates the difference in resilience becomes much larger. We have added clarification in the revised version to make this point explicit.
>
> **Training runtimes and model sizes:**
> We have added training times for the different precision levels and model sizes in terms of number of bits in the appendix, as requested.
>
> **Practical significance of the symmetry results:**
> We agree that corruption rates near p=1 are not representative of normal hardware operating conditions, and we do not intend the even-layer recovery result to be interpreted as a practical deployment claim for such BERs. Rather, we view it as an interesting theoretical stress-test phenomenon that reveals a nontrivial cancellation symmetry unique to logic/LUT-based networks. Its main value is conceptual: it suggests that anti-symmetric LUT structures may expose fault-cancellation mechanisms that could be useful for future work in information theory, coding theory, and resilient logic-based architectures. In particular, such mechanisms may eventually help design methods that achieve even greater resilience at lower, more practical BER regimes. We have clarified in the revised manuscript that this result is primarily a theoretical discovery, not a claim of immediate practical utility.

---

### Review · Reviewer_ZRjT · 2026-04-21

**Summary Of Contributions:**

The paper studies the resilience of neural networks to independent, random bit-flips in parameter memory. The central idea (as authors state) is that resilience should be analyzed as a *structural property* of the architecture rather than as a property of a particular trained solution. The claimed contributions are -- (i) Closed form error expressions under random bit-flips, (ii) A clean reslience hierarchy (iii) Analysis of other factors such as activation, width, sparsity and depth on error (iv) LUT networks as a final optimum design (v) Empirical validation on MLPerf.

**Strengths:** I liked the reframing of analysis as "structural" and analysis as "resilience of architecture". The theory is decently well laid out and the intuition behind the "resilience hiearchy" follows from the theory. To the extent I could verify, the statements and proof are correct.

**Weaknesses:**

- One issue I had was -- Each subsection seemed to consider a different models/abstracttons for analysing errors. And it's not obvious if all these models are compatible/consistent with each other. -- a generic noise $\xi$; a bias-variance decomposition with unspecified $\Delta w$ statistics; a uniform Lipschitz gain $\lambda$ with an unspecified intrinsic $\nu$ --. The sections should be chained into a single end-to-end with common assumptions.

- In practice what we really care about is not the error caused by bit-flips on train, but (generalization) error which propagates to test. The analysis, as it seems, focusses on error caused by bit-flips w.r.t trained weights which will give bounds on train accuracy. But, one should discuss the generalization errors as well, if not theoretically then atleast empirically.

- I was unable to understand which figures are the CNN baselines on MLPerf from the captions or the text. This should be pointed out explicitly.

**Additional Comments:**

NA

**Audience:**

Yes

**Audience Explanation:**

**Yes.** The paper sits at the intersection of hardware-aware ML, quantization, and logic/LUT-based edge inference all of which have been active at TMLR and allied venues. The structural framing of resilience is a contribution that researchers in efficient inference, and hardware–software co-design would find relevant.

**Claims And Evidence:**

No

**Claims Explanation:**

Actually, my answer is -- Yes, with required clarifications.

The core theoretical claims are derived rigorously (to the extent I could verify) and match the empirical trends in Figures.The empirical sweeps are broad and consistent with the theory.

However, two claims currently outrun the evidence:
1. The claim of "Accuracy–Resilience trade-off" is never well evidenced nor proved.
2. Claim "Same resilience trends appear in representative edge architectures (CNNs)"* has no supporting results as it's written now..

Please also see the **Weaknesses** above.

**Requested Changes:**

1. The CNN baseline results need to be either shown or the claim removed. Section 5.1 states that the authors "additionally report results against the standard MLPerf Tiny CNN baselines", but I was unable to locate these results anywhere.

2. The ablations in section 5.3 should specify under which paradigm are these performed. Ideally, one needs both a baseline (clean full precision) as well as LUT on the same figure so that comparisons are easy.

3. The theorems compute the squared drift between clean and corrupted outputs on the same input,  which is not the same as a bound on test accuracy. The narrative treats the two interchangeably. Some discussion of generalization error, at least empirically, would strengthen the paper.

4. Each of subsections in 4 introduces its own abstraction for the error model - a generic noise, a bias-variance decomposition, a uniform Lipschitz gain $\lambda$ with an unspecified $\nu$ - and it's not clear that these are mutually consistent. Either a unification or an explicit acknowledgement that "the four structural factors are analyzed under modular independent abstractions with end-to-end composition left to future work" is needed.

---

> ### Author Response · Authors · 2026-05-17
>
> We thank the reviewer for the detailed feedback and for recognizing the structural framing and theoretical contributions of the paper.
>
> **CNN baseline results:**
> We thank the reviewer for pointing out the missing CNN results. In the previous version, Figure 7 mistakenly contained the MLP result instead;. This has been updated to include the CNN results. We thank the reviewer for identifying this oversight. As seen in the updated figure 7, the resilience trend remains the same, as predicted by our theory. The accuracy-resiliency trade-off is also present there, with higher precision achieving higher initial accuracy, but dropping faster when bit corruption increases.
>
> **Accuracy–resilience trade-off:**
>  This effect is directly shown in all ablation experiments. For example, in Figure 1, the clean-accuracy differences may visually appear almost the same because they are small compared with the much larger drops induced by corruption, but they are still present. In KWS, FP32 starts at approximately 78% accuracy, while the BNN starts at approximately 74%. However, at p=0.2p = 0.2p=0.2, FP32 drops to approximately 11%, whereas the BNN remains around 72%, showcasing the trade-off. A more visually evident example appears in the sparsity plots in Figure 6: clean accuracy is much higher at S=0S = 0S=0 than at S=0.95S = 0.95S=0.95, but the dense model’s accuracy drops much faster as corruption increases.
>
> **Clarification of ablation baseline:**
> The baseline is already present and is the same across all ablation studies: each ablation starts from the same FP32 one-hidden-layer fully connected MLP baseline, with 256 hidden neurons and ReLU activation, and only the specific hyperparameter under study is varied. We have added this information explicitly in the revised version to make the experimental setup clearer.
>
> **Accuracy interchangeability:**
> We would like to clarify that we do not use accuracy interchangeably with output error in the theoretical development. The theoretical results are stated in terms of corruption-induced output error, specifically the expected squared drift between the clean and corrupted outputs for the same input, since this is the quantity being proven under the bit-flip model.
>
> Accuracy is mentioned only in the experimental evaluation, where it is used as an empirical downstream metric to show how output corruption affects benchmark performance. The only theoretical discussion motivated by accuracy is the LUT symmetry/recovery analysis, which is used only to motivate and narrate the empirical recovery behavior observed in the experiments, but is never used in the theorems, corollaries and proofs, being consistent with the scope of the paper.
>
> **Accuracy bounds / generalization:**
> Generalization under corruption is a different problem, closer to generalization theory and VC-style analyses, and is not the goal of this paper. Our goal is to study bit-flip resilience by analyzing how memory-level parameter corruption affects model outputs as a function of architectural structure, independently of how these output errors may affect generalization.
>
> This distinction is important because output corruption is the direct consequence of hardware bit flips and can be analyzed in a dataset-agnostic and metric-agnostic manner, whereas generalization depends on the data distribution, task, loss, training procedure, and decision boundary.
>
> **Consistency of theoretical abstractions:**
> Each theoretical subsection introduces assumptions tailored to the architectural component being analyzed. This modular structure is intentional: an assumption-free exact analysis of arbitrary full-network corruption is generally intractable and depends exponentially on the network topology and parameter states. The purpose of the theory is to isolate structural factors—precision, activation boundedness, width, sparsity, and depth—and show how each one affects expected corruption-induced error under a controlled abstraction. The experiments then validate that the trends predicted by these modular analyses are observed empirically

---

### Review · Reviewer_aVfW · 2026-05-04

**Summary Of Contributions:**

This paper presents a theoretical and empirical analysis of neural network resilience to hardware bit-flips. It successfully demonstrates that Logic and Lookup-Based Neural Networks (LUT-NNs) offer superior robustness due to their fundamental structural properties.

**Strengths** :

1. The paper provides rigorous mathematical derivations for the Expected Squared Error under parameter bit-flip corruption across multiple numerical formats and architectural configurations.
2. It also identifies and formalizes a novel "Even-Layer Recovery" phenomenon that occurs uniquely in LUT models under extreme fault rates
3. The empirical results on edge computing benchmarks strongly align with the theoretical predictions regarding precision, activation bounds, and sparsity.

**Weaknessess**:

1. The papers' ablation studies primarily tackle MLPs, which question the generalizability of the theory.

**Additional Comments:**

As this paper is more or less beyond my research expertise, I would rate the confidence of my assessment of this paper as 2/5. I would request the AE to please take this confidence score into account when deciding on this paper.

**Audience:**

Yes

**Audience Explanation:**

This research is highly relevant to the machine learning community focused on edge computing, hardware-software co-design, and system reliability. Anyone developing low-power edge accelerators will find the formalization of lookup-table robustness particularly valuable for future architectural designs.

**Claims And Evidence:**

Yes

**Claims Explanation:**

The authors claim that lower precision, higher sparsity, bounded activations, and shallow depths inherently improve fault resilience. They support this with a dataset-agnostic mathematical framework contrasting the unbounded multiplicative errors of floating-point representations against the localized errors of LUT-based neurons. Empirical evaluations on the MLPerf Tiny benchmark validate these claims by demonstrating that standard floating-point models collapse at low error rates, whereas LUT-based models maintain stability significantly deeper into the corruption regime.

**Requested Changes:**

Please provide comments on the generalisability of the theory and experiments on newer models, particularly unimodal and multimodal foundation models.

---

> ### Author Response · Authors · 2026-05-17
>
> We thank the reviewer for the positive assessment and for highlighting the relevance of the theoretical derivations, the empirical validation, and the even-layer recovery phenomenon.
>
> **Comments on the generalisability of the theory and experiments on newer models, particularly unimodal and multimodal foundation models:** The theory is not specific to MLPs, it is stated at the level of architectural components such as numerical representation, bounded activations, sparsity, fan-in, and depth, which also appear in CNNs, transformers, and modern unimodal and multimodal foundation models. Therefore, the theorems derived in the paper apply directly to newer architectures as well, although the magnitude of the effect can vary. The MLP ablations are used because they allow controlled isolation of each architectural factor while running many corruption rates and Monte Carlo trials; performing the same exhaustive sweep on modern foundation models would require substantially more compute than available for this study. This choice limits only the scope of empirical validation, not the generality of the theory. The revised CNN results further show that the same qualitative trends appear beyond the controlled MLP setting in representative MLPerf Tiny edge architectures.

---

> > ### Comment · Reviewer_aVfW · 2026-05-19
> >
> > Thank you for your response! My issues are resolved.

---

### Public Comment · ~Chuanjian_Liu1 · 2026-04-03
**Decline of Review Invitation Due to Research Focus Misalignment**

Thank you for inviting me to review this manuscript. However, my current research focus is not aligned with the paper's topic. I regret that I cannot provide a thorough evaluation. Please consider finding a specialist in this specific field.

---

### Decision · Action_Editor_7ukw · 2026-06-03

**Recommendation:** Accept with minor revision

**Additional Comments:**

The authors should take another pass through the paper to make sure they have addressed all reviewer comments in their revised manuscript

Some points that I think would deserve some more thorough discussion in the paper:
1. Do the authors believe the assumption of independent bit flips is realistic? Often in hardware we may observe corruption that is not independent (for example if an entire memory cell stops functioning properly). I suggest a discussion is added to clarify how the results may be limited by non-independent errors.

2. Equation 22: Clarify more what this error represents in the accompanying text

3. Many references such as in Section 4.8.1 do not seem to be encompassed in parentheses. e.g.: "for inference Umuroglu et al. 2020...." Double checks this

4. While briefly discussed in the rebuttal, I would have appreciated a more thorough discussion on learnability of such LUT models (e.g., VC-dimensions, Rademacher complexities etc) and how this may limit their conclusions

Typo in Section 4: "Section ??"

**Audience:**

Yes

**Audience Explanation:**

The reviewers seem to agree that this research is highly relevant to the machine learning community focused on edge computing, hardware-software co-design, and system reliability. Researchers in efficient inference, and hardware–software co-design would find it relevant.

**Claims And Evidence:**

Yes

**Claims Explanation:**

There is agreement amongst the reviewers that the authors' claim that lower precision, higher sparsity, bounded activations, and shallow depths, inherently improve fault resilience. The reviewers seem to be mostly in agreement that these claims are supported by the tests performed and that the empirical sweeps are broad enough.

Some reviewers had expressed some concerns such as that the claim of "Accuracy–Resilience trade-off" is never well evidenced nor proved, that some claims have no supporting results. One reviewer also expressed concern that the computational cost may have been high. However in their rebuttal/revision the authors seem to have addressed these concerns to the satisfaction of the reviewers. In their final submission the authors should make sure they have addressed the requested changes as the reviewers accepted the paper based on these rebuttal responses/revisions.

---

> ### Author Response · Authors · 2026-07-01
>
> We thank the Action Editor for the positive decision and constructive minor-revision requests.
>
> In the revised manuscript, we made the following changes as requested:
>
> 1. Added a discussion of the independent-bit-flip assumption, including when it is appropriate and how correlated hardware faults such as burst errors, stuck-at cells, and word-line failures limit the exact MSE expressions ("Scope of the independent-bit model" paragraph in section 4.1)
>
> 2. Clarified Eq. 22. We now state explicitly that this quantity is not an output-error formula, but the probability/fraction that a random corrupted bit lands in the shared affine quantization metadata rather than an ordinary integer weight.
>
> 3. Fixed citation formatting throughout the manuscript.
>
> 4. Added a new Background paragraph on the expressivity, efficiency, and scalability motivation for logic/LUT-based networks. This paragraph cites the VC-dimension analysis of 1-layer LUT-based models, notes that LUT compositions form a high-dimensional discrete function-approximation substrate, and clarifies that recent results mainly demonstrate strong efficiency for MLP-style or hybrid LUT/weighted architectures rather than fully replacing large CNNs or Transformers end-to-end. We also clarify the motivation of our work in this context: logic/LUT-based networks are promising not only because they can provide extremely low-energy, low-latency, and high-throughput inference, but also because they provide extreme structural resilience to parameter bit flips. This motivates further research into scaling these models toward fully convolutional, transformer, and foundation-model settings.
>
>   Fixed the “Section ??” typo by labeling the Depth subsection and referencing it properly.